# Development and validation of an asthma self-knowledge questionnaire

**Adalberto Fernandes Dos Santos**[1,2,3], **Renata Costa**[1], **Henrique Pereira**[1,4,5], **Ana Rita Pedro**[6,7], **Luis Taborda-Barata**[1,3,8]*

**1** RISE-Health, Faculty of Health Sciences, University of Beira Interior, Covilhã, Portugal, **2** Faculty of Medicine, Universidade Agostinho Neto, Luanda, Angola, **3** UBIAir—Clinical and Experimental Lung Centre, UBIMedical, University of Beira Interior, Covilhã, Portugal, **4** Faculty of Social and Human Sciences, University of Beira Interior, Covilhã, Portugal, **5** The Research Center in Sports Sciences, Health Sciences and Human Development (CIDESD), Covilhã, Portugal, **6** NOVA National School of Public Health, Public Health Research Center, NOVA University Lisbon, Lisbon, Portugal, **7** Comprehensive Health Research Center, NOVA University Lisbon, Lisbon, Portugal, **8** Immunoallergology Department, Cova da Beira Local Health Unit (ULS), Covilhã, Portugal

* tabordabarata@fcsaude.ubi.pt

## Abstract

### Background

Validated questionnaires for adequately assessing knowledge about asthma are scarce. Thus, the primary objective of the present study was to develop and validate an asthma self-knowledge questionnaire, based on international recommendations on the disease. The secondary objectives were to compare knowledge about asthma between asthmatic patients and non-asthmatic individuals; assessing whether asthma affects the level of self-knowledge of the disease and what factors may be associated with poorer self-knowledge of the disease.

### Methods

The Bronchial Asthma Self-Knowledge Questionnaire was developed, and validation studies were performed: logical or apparent validity, content validity, construct validity; internal consistency (Cronbach's alpha test), test-retest or reproducibility, in a face-to-face survey with 104 asthmatic patients and 131 non-asthmatic individuals (n = 235). Other questionnaires were also applied: Mini Mental State Examination (in individuals over 65 years of age), Depression Scales (CES-D for individuals under 65 and GDS for individuals over 65 years of age), Demographic Questionnaire, Health Literacy, and the Characterization Questionnaire for bronchial asthma.

### Results

Regarding development of the questionnaire, content validity, determined using I-CVI allowed reducing the questionnaire to 21 items. The test proved to have an acceptable value of in-ternal consistency and the data were considered as normally

**Data availability statement:** The data underlying the results presented in the study are available from Zenodo (https://zenodo.org/); the identifier is 10.5281/zenodo.11155084.

**Funding:** Adalberto dos Santos was awarded a PhD grant (Reference: 2022.11223.BD) from the Portuguese Foundation for Science and Technology (Fundação Portuguesa para a Ciência e Tecnologia). The funders had no role in study design, data collection and analysis, decision to publish, or preparation of the manuscript.

**Competing interests:** The authors have declared that no competing interests exist.

distributed; the questionnaire presented good temporal stability, by test-retest, although Spearman rho values were significantly stronger in the asthmatic group. Finally, confirmatory factorial analysis yielded acceptable values for PCFI and PGFI, as well as a satisfactory value for RMSEA. In terms of the application of the questionnaire, both groups under study (asthmatics and non-asthmatics) showed statistically significant differences in replies of self-knowledge questionnaire items. Finally, factors such as health literacy disturbances seem to marginally influence self-knowledge of bronchial asthma.

## Conclusions

The developed and validated questionnaire showed adequate psychometric robustness. In terms of construct validity, by known group (bronchial asthma) validity, the test was able to discriminate between patients with asthma and participants without asthma, regarding self-knowledge of the disease.

---

## Introduction

Asthma is a heterogeneous disease, usually characterized by chronic bronchial inflammation and may be associated with bronchial hyperreactivity due to various stimuli, such as allergens or physical exercise. It is defined by the presence of obstruction to pulmonary airflow, which is reversible spontaneously or with appropriate treatment. [1] It is one of the most common chronic diseases worldwide and it is estimated to currently affect around 262 (95% CI: 224–309) million people worldwide, with a high burden of disease [2]

In Portugal, an epidemiological study of 2012 detected 695,000 Portuguese with asthma, suggesting a prevalence of 6.8%. [3] In clinical terms, in cases where asthma symptoms are more persistent, it is extremely important that these symptoms can be controlled through pharmacological and nonpharmacological measures. Overall, in these cases, it is essential that regular and adequate medication is carried out and that there is a good plan for asthma monitoring and self-management [1]. However, for this approach to succeed, it is very important that asthmatic patients know their disease, to identify and prevent exacerbations, and to know how to adjust the treatment.

In fact, various studies have shown, in different settings, that health literacy is closely related to health outcomes. In this context, Pedro AR et al stated that "inadequate health literacy (when compared with adequate health literacy) is strongly linked to poor knowledge or understanding of both care delivery services and health outcomes and may also be associated with (...) a high prevalence and severity of some chronic diseases... " [4]. In terms of asthma, there are also studies showing that increasing health literacy and, consequently, increasing knowledge about bronchial asthma leads to better outcomes in relation to disease control, because patients understand better the importance of proper treatment and the best time to start it, as part of a self-monitoring approach, guided by the doctors who follow them [5–7].

Although there are bronchial asthma self-knowledge questionnaires that have already been validated, in several countries, there is some heterogeneity across questionnaires [7–10], which has been reviewed by Pink et al [11]. Furthermore, several of these questionnaires are old and have outdated asthma concepts, and lack some additional aspects regarding asthma, which must be included to have adequate and complete tools to evaluate self-knowledge about the disease.

A knowledge questionnaire on bronchial asthma is an instrument that can help to detect knowledge gaps in patients, which should be addressed. It is, therefore, important that a questionnaire that analyses the knowledge gap of each patient be developed and validated in an appropriate way. To do this, it will be important to have the notion that a valid and high-quality knowledge assessment questionnaire should have good discriminative and evaluative properties and should be able to detect small differences between patients with different levels of knowledge [12].

In Portugal, although a general questionnaire on health literacy, the European Health Literacy Survey [4], has already been translated and validated, to the best of our knowledge, no bronchial asthma self-knowledge questionnaire has yet been validated. Thus, the primary objective of this study was to develop and validate a questionnaire for self-knowledge of bronchial asthma based on international recommendations on the disease. The secondary objectives of the study were to compare knowledge about asthma between asthmatic patients and non-asthmatic individuals and to analyze which factors might influence self-knowledge of the disease.

## Materials and methods

### Questionnaire development

A Bronchial Asthma Self-Knowledge Questionnaire was developed based on international recommendations and on the Global Initiative for Asthma (GINA) [1], as well as on similar questionnaires. Regarding item generation procedures, we employed a deductive method approach to classify typologies of items, based upon prior literature, theoretical definitions, and guidelines. Writing of survey items followed several rules to ensure that they were properly constructed, such as: items addressed one single issue (double-barreled items were strongly avoided), were written consistently in terms of perspective (we avoided mixing items related to information, or attitudes), we used statements that were simple and as short as possible with familiar language to target respondents, and avoided negatively-worded or reverse-scored, since this can have a detrimental effect on the comprehension of its content. Finally, we checked for redundancies and pretested items for content adequacy. All questions were proposed by the authors and assembled into this novel questionnaire.

Categorization of self-knowledge scores was done using a 5-point Likert Scale which offered five different options for the respondents to choose from. The options included two extremes (1/5), two intermediate (2/4), and one neutral opinion (3). For the purpose of this research, data were analyzed by using mean scores, calculated for each statement in the Likert scale. Hence, self-knowledge was then determined based on the argument that a mean score of 3 in Likert scale represents neutral knowledge, mean score of less than 3 represents negative or lack of knowledge and greater than 3 represents a positive and accurate knowledge.

### Theoretical construct analysis

**Logical or apparent validity.** The initial questionnaire was reviewed by three Immunoallergology specialists, to evaluate its logical or apparent validity, namely in terms of intelligibility and logic. International guidelines and consensus documents were followed. A pilot study was also conducted in 10 asthmatic patients and 10 healthy adult controls to assess the comprehension and adequacy of the terms used.

**Content validity.** The base questionnaire was then reviewed by seventeen Immunoallergology experts, who classified each questionnaire item/ question in terms of relevance regarding current guidelines and scientific knowledge about asthma, according to the following parameters used to evaluate the content: 1- not relevant; 2-something relevant; 3-very relevant; 4-extremely relevant [13]. The I-CVI (Item Content Validity Index) was then calculated based on the number of

experts who assigned a rating of 3 or 4, divided by the total number of experts. I-CVI is considered significant if its value is equal to or greater than 0.78 [14]. Questions with I-CVI below this value were modified or deleted.

**Empirical construct analysis: Construct validity.** Construct validity was assessed by known group validity [15,16]. Thus, a pilot study was conducted in 40 patients with clinically confirmed and characterized bronchial asthma, and 40 healthy controls, all adults, to evaluate the capacity of the questionnaire to detect knowledge differences between individuals with and without asthma.

**Sensitivity analysis.** Sensitivity analysis of the items allows assessment of the error present in the evaluation of each item and was studied by the measures of asymmetry (Skewness) and flatness (Kurtosis) according to their respective critical ratios. Items with absolute values of skewness higher than 3 or with absolute values of kurtosis higher than 7 were eliminated since this would describe insensitive distribution parameters [17].

## Reliability analysis

**Internal consistency.** The questionnaire reliability was evaluated by studying the questionnaire's internal consistency, that is, the degree to which the various questions assess the same aspects. Thus, the second pilot study in 40 patients with clinically confirmed and characterized bronchial asthma and 40 healthy controls, all adults, also served to evaluate this component by calculating Cronbach's alpha value [15,18].

**Test-retest.** The reliability of the questionnaire was also evaluated by re-applying the same questionnaire to 23 patients with asthma and to 20 healthy volunteers, with an ideal interval of two weeks, which allowed to evaluate the stability of the answers to each question. This reapplication allowed us to calculate the intraclass coefficient of each question, between the two applications of the questionnaire [15,19].

## Questionnaire application

The bronchial asthma self-knowledge questionnaire was later applied in a face-to-face interview to 73 asthmatic patients and 76 non-asthmatic individuals from the following hospitals: Cova da Beira University Hospital Centre (Covilhã), Covilhã Health Centre, Tondela-Viseu Hospital Centre (Viseu) and Hospital Amato Lusitano (Castelo Branco), which, together with the University of Beira Interior, are part of CACB – Clinical Academic Center of Beiras.

## Confirmatory factor analysis

Confirmatory Factor Analysis (CFA) evaluated the questionnaire's factor structure. This choice was guided by the instrument's theoretical underpinnings: each item was developed from established constructs and supported by prior literature, providing a clear a priori model to test. In situations where a predefined structure is well supported, CFA is generally preferred over Exploratory Factor Analysis (EFA), as it allows for direct testing of hypothesized relationships between observed variables and latent constructs [20].

Before carrying out the CFA, we examined the adequacy of the data for factor analysis. The Kaiser–Meyer–Olkin (KMO) statistic exceeded the recommended threshold, and Bartlett's test of sphericity was significant, confirming that the correlation matrix was suitable for further analysis. Although the questionnaire is being applied in a new context, its strong theoretical foundation and supportive diagnostic results justified proceeding directly with CFA rather than conducting a full EFA. This approach is consistent with prior recommendations that instruments grounded in robust theory and empirical evidence may bypass EFA and move directly to CFA [21]. These steps helped confirm that CFA was an appropriate next step for testing our hypothesized model. The CFA is a multivariate statistical procedure that tests how a given number of items represent a few dimensions or factors. We specified the number of dimensions or factors with latent variables to which the corresponding items were related: General Aspects (Q1, Q2, Q3, Q4, Q5, Q6); Pathophysiological/ Clinical Aspects (Q7, Q8, Q9, Q10, Q11, Q12); Therapeutic Aspects (Q13, Q14, Q15, Q16, Q17, Q18); Non-pharmacological Therapeutic Aspects (Q19, Q20, Q21). Several indices of quality of adjustment were used to confirm the factorial validity,

namely Chi2 (Model Chi-Square), CFI (Comparative Fit Index), PCFI (Parcimonious Bentler's Comparative Fit Index), GFI (Fit Index), the PGFI (Parcimonious Goodness-of-Fit Index) and RMSEA (Root Mean Square Error of Approximation), using the AMOS program [22].

## Other questionnaires

Other questionnaires were also applied, already translated, and validated in Portuguese: Mini Mental State Examination, which allows distinction between cognitive deficit and no cognitive deficit [23,24]; the Geriatric Depression Scale – GDS, a 15-item questionnaire used in individuals older than 65 years, and which allows classification into not depressed, mildly depressed, and severely depressed states [25], as well as the Depression Scale (CES-D), a 20-item questionnaire used in non-elderly individuals to classify them into not depressive or depressive [26]; Demographic Questionnaire; Mini-questionnaires on asthma severity based on medication [1,27], asthma control [1,27,28] and also a list for recording medication used in asthma (in patients with asthma), as well as other morbidities (in patients with asthma and in volunteers without respiratory diseases); and the European Health Literacy Survey [1,29]. Authorisations were obtained to use these questionnaires. These questionnaires were used to characterize the samples of patients and non-asthmatic controls, and also to ascertain that eventual differences found between the two groups of volunteers regarding asthma literacy would not be due to underlying differences in depression or other psychological or sociodemographic features.

## Selection of asthmatic and non-asthmatic patients

For both the pilot study and the questionnaire application study, patients with asthma and non-asthmatic volunteers were selected based on convenience samples. Recruitment period took place in two periods, between 7th January and 27th February 2019, and also between 14th March and 17th June 2022. Selection criteria were as follows:

**Asthmatic patients.** Inclusion criteria: i) To be 18 years of age or older. ii) Having medically proven asthma. iii) Having regular follow-up in consultations, due to asthma. Exclusion Criteria: i) Inability to understand the objectives of the study/ cognitive deficit. ii) Having respiratory diseases other than asthma.

**Non-asthmatic participants.** Inclusion criteria: i) To be 18 years of age or older. ii) Having regular follow-up in hospital consultations. Exclusion Criteria: i) Inability to understand the objectives of the study/ cognitive deficit. ii) Having respiratory diseases (asthma or other).

## Sample size calculation

The sample size calculation was based on a similar previous study performed in New Zealand [9], which also involved an asthma knowledge questionnaire, applied in patients with asthma and volunteers without asthma. Considering the results of this study, and to find a difference greater than 2.5 points in the average scores of the asthma knowledge questionnaire, between the two groups under study, with a power of 0.80, an error of no more than 2% and a test value of less than 5%, we needed to test 90 individuals in each group. Considering an adhesion rate of 50%, it was necessary to invite 180 individuals in each group, which we decided to adjust to 200 in each. It should be borne in mind that this sample size calculation was based on the capacity of the questionnaire to detect differences between the two groups and a such a high number of volunteers was not needed to test the basic features of the questionnaire, in terms of its validation. This calculation was applied to both the development and validation of the asthma self-knowledge questionnaire, and the knowledge comparison between asthmatic and non-asthmatic participants.

## Statistical analysis

Results were statistically analyzed using SPSS software, version 27. Descriptive analysis was used to characterize the sample and Mann-Whitney, or Student's t test were used to compare results between the two study groups (asthmatics versus non-asthmatics). The Pearson Chi-square test was used to analyze the influence of the asthma self-knowledge

score by quartiles on asthma control. The I-CVI was calculated in percentage terms. Cronbach's alpha test was used to evaluate the reliability of the questionnaire in terms of internal consistency. Spearman's Rho correlation coefficient was calculated and values greater than 0.70 were regarded as having a strong correlation. Intraclass Correlation Coefficient of each question was calculated to evaluate the reliability of each question regarding test-retest (stability), and the obtained levels were interpreted in terms of agreement: 0.00 – weak; 0.01-0.20 – slight; 0.21-0.40 – sufficient; 0.41-0.60 – moderate; 0.61-0.80 – substantial; >0,80 – almost perfect [30]. The tests whose respective test value (p) did not exceed 0.05 were considered significant.

### Ethical approval

Approval was obtained from the Ethics Committee of the University of Beira Interior (CE-UBI-Pj-2018-070), as well as implicitly from the Ethics Committees of the hospitals involved in the study, since UBI and these clinical institutions are all part of the same Clinical Academic Center – CACB – Clinical Academic Center of Beiras. All asthmatic patients and volunteers without asthma who agreed to participate in the study signed a written, free and informed consent, after having been clarified any doubts that might arise. All rights of patients and volunteers were respected under the Helsinki Declaration and its updates.

### Data protection

All the appropriate procedures provided for in the Portuguese legislation on data protection (Law nº67/ 98; Regulation (EU) No. 2016/679 and its Rectification) were followed in terms of data collection and storage. Data collection forms were pseudo-anonymized. Likewise, the disclosure of results will always be anonymized, and no personal data will come from each non-asthmatic patient or volunteer. Only the principal investigator had access to the data decryption key. Originals of each questionnaire were stored in a locked location and the data transferred to a computer base is in a file with encrypted access.

## Results

### Content validity—I-CVI

Seventeen Immunoallergology experts classified each questionnaire item/ question in terms of relevance to current guidelines and scientific knowledge about asthma. The initial questionnaire had 24 questions, but questions 13, 20 and 21 were deleted because their I-CVI value was below 0.78 (Table 1).

### Sensitivity analysis

Questionnaire items were then studied in terms of sensitivity, through analyses of asymmetry (Skewness) and flattening (Kurtosis). No item had absolute values of skewness higher than 3. Just one item, Q3, had an absolute value of kurtosis higher than 7 (9.752) but it was not eliminated due to the fact that it was considered relevant by the authors. Results are shown in Table 2.

### Internal consistency

The reliability analysis of the overall 21-item of the Asthma Self-knowledge Questionnaire returned a Cronbach's alpha coefficient of 0.656. If we removed question 8, Cronbach's alpha coefficient would be 0.725, but we decided to keep that question because its removal would not have a significant effect.

### Test-retest

We calculated the intraclass correlation coefficient (two-way mixed model, absolute agreement type) to evaluate the reliability of each question regarding test-retest (stability); results are shown in Table 3. Regarding the average intraclass

**Table 1. Initial questionnaire developed for the study, and parameters used for calculating I-CVI (Item Content Validity Index).**

| Question number | Examples of questions | Number of experts rating item as 3 or 4 | Total number of experts | I-CVI |
|---|---|---|---|---|
| Q1 | Asthma is a chronic disease which persists even during periods without symptoms. | 17 | 17 | 1.00 |
| Q2 | Asthma begins more frequently in children or young adults than in the elderly. | 14 | 17 | 0.82 |
| Q3 | With good medical follow-up, most asthmatics can lead a normal life. | 17 | 17 | 1.00 |
| Q4 | When asthma is not treated, it is a disease that may kill. | 15 | 17 | 0.88 |
| Q5 | In an asthmatic patient, an episode of flu may trigger an asthma bout. | 16 | 17 | 0.94 |
| Q6 | People with allergies are more likely to have asthma than people without allergies. | 16 | 17 | 0.94 |
| Q7 | The airways of the lungs (bronchi) are inflamed in asthma. | 16 | 17 | 0.94 |
| Q8 | The more inflamed bronchial airways are, the more severe is asthma likely to be. | 14 | 17 | 0.82 |
| Q9 | Some asthma symptoms are due to narrowing of the bronchi (lung airways). | 15 | 17 | 0.88 |
| Q10 | Coughing frequently may be a symptom of asthma. | 17 | 17 | 1.00 |
| Q11 | Asthma causes episodes of shortness of breath, but these are not really dangerous. | 14 | 17 | 0.82 |
| Q12 | An asthmatic patient should always go to an Emergency Department whenever he/she has mild shortness of breath. | 14 | 17 | 0.82 |
| Q13* | The diagnosis of asthma can be confirmed by Chest X-ray. | 8 | 17 | 0.47 |
| Q14 | An asthmatic patient who needs to use his/her rescue inhaler (p.r.n.) for asthma bouts many times weekly, has his/her asthma controlled. | 14 | 17 | 0.82 |
| Q15 | Asthma can be well controlled in terms of symptoms. | 16 | 17 | 0.94 |
| Q16 | An asthma bout may be resolved by taking an anti-allergic pill. | 14 | 17 | 0.82 |
| Q17 | Hand tremor can be a side effect of rescue medication used in asthma bouts. | 14 | 17 | 0.82 |
| Q18 | An asthmatic patient should hold his/her breath for 10 seconds after each inhaler medication inhalation. | 14 | 17 | 0.82 |
| Q19 | Asthmatic patients do not need to take medication outside of asthma bouts. | 14 | 17 | 0.82 |
| Q20* | Aspirin may trigger an asthma bout. | 8 | 17 | 0.47 |
| Q21* | The bedroom of an asthmatic patient should always be closed. | 6 | 17 | 0.35 |
| Q22 | Avoiding exposure to situations that may trigger an asthma bout, such as tobacco smoke and house dust, may improve asthma control. | 14 | 17 | 0.82 |
| Q23 | Asthmatic patients should avoid doing any type of physical exercise. | 14 | 17 | 0.82 |
| Q24 | Certain sports, such as swimming, are better for asthmatic patients. | 14 | 17 | 0.82 |

(*) Items excluded from the questionnaire due to having low I-CVI.

correlations scores, they were significantly stronger in the asthmatic group compared to non-asthmatic group. In the asthmatic group, questions 1, 3, 4, 5, 7, 8, 9, 10, 12, 13, 15, 17, 18, 19, 20 and 21 had strong and statistically significant correlations with their respective re-test; while in non-asthmatics there were only a strong correlation between question 4, 5, 6, 7, and 10 and their re-test.

## Sample selection

For the present study, patients were recruited from CHUCB, UCSP Covilhã, ULS of Castelo Branco and CHTV. Most asthmatic patients were recruited at CHUCB and most non-asthmatic volunteers were recruited both at CHUCB and at UCSP Covilhã.

Two hundred and thirty-five (235) people accepted to participate in the study, and 104 were asthmatic and 131 were non asthmatic. All of them met the inclusion criteria in the study, and we did not have to exclude any patient (Fig 1).

**Table 2. Sensitivity analysis.**

| Items | Kurtosis | Skewness |
|---|---|---|
| Q1 | −1.416 | 1.376 |
| Q2 | −0.660 | −0.419 |
| Q3 | −2.832 | 9.752 |
| Q4 | −1.393 | 1.099 |
| Q5 | −1.181 | 1.161 |
| Q6 | −1.150 | 0.852 |
| Q7 | −0.998 | 0.516 |
| Q8 | 0.342 | −1.280 |
| Q9 | −0.970 | 0.504 |
| Q10 | −0.460 | −0.678 |
| Q11 | 0.049 | −1.526 |
| Q12 | −0.043 | −1.299 |
| Q13 | −0.067 | −1.565 |
| Q14 | −1.182 | 1.269 |
| Q15 | −0.282 | −1.281 |
| Q16 | −0.035 | 0.952 |
| Q17 | −0.614 | −0.327 |
| Q18 | −0.238 | −1.382 |
| Q19 | −2.287 | 5.638 |
| Q20 | −0.345 | −1.444 |
| Q21 | −0.696 | 0.000 |

## Sociodemographic results

Sociodemographic characteristics of the studied population are shown in Table 4.

The mean of age of all 235 participants was of 45.27 ± 15.85 (mean+SD) years. Asthmatic and non-asthmatic groups were well matched in terms of age (44.26 ± 15.98 versus 47.87 ± 15.62 years, respectively). The two groups were also in concordance in terms of gender, with a clear predominance of women, urban residence, and in terms of occupation, most participants were employed (65% in the asthmatic group and 64.9% in the non-asthmatic volunteers' group). In terms of education, more participants had university education (38.5%) in the asthmatic group than in the non-asthmatic group (33.6%).

Regarding whether there was anyone in the family with asthma, there were significant differences between the two groups, as expected: in the asthmatic group, twice as many people had a relative with asthma, when compared with the non-asthmatic group (p < 0.001; Chi-square test). There was also a discrepancy between the two groups in relation to contact with asthmatic persons: in the asthmatic group twice as many people had contact with other asthmatic patients, when compared with the non-asthmatic group (p < 0.001; Chi-square test).

None of the patients had cognitive impairment since it was an exclusion criterium. Most of the elderly people in the two groups (asthmatic and non-asthmatic) had no depressive humor on the GDS, as was the case in CES-D, where most people had normal humor.

## Clinical characterization

Clinical characteristics of the patient sample are shown in Table 5. Most asthmatic participants had asthma for about 1–10 years (44.1%) and a small minority (3.2%) had asthma for more than 41 years (Mean = 16.18; SD = 12.10). More than half

**Table 3. Test-retest analysis.**

|  | Average Intraclass Correlation (Asthmatics) | p value | Average Intraclass Correlation (Non-asthmatics) | p value |
|---|---|---|---|---|
| Q1*RT1 | 0.617 | 0.009* | 0.055 | 0.444 |
| Q2*RT2 | −0.726 | 0.885 | 0.516 | 0.068 |
| Q3*RT3 | 0.756 | 0.000** | −0.565 | 0.818 |
| Q4*RT4 | 0.623 | 0.015* | 0.593 | 0.005* |
| Q5*RT5 | 0.621 | 0.015* | 0.622 | 0.023* |
| Q6*RT6 | 0.465 | 0.074 | 0.552 | 0.049* |
| Q7*RT7 | 0.707 | 0.003* | 0.612 | 0.018* |
| Q8*RT8 | 0.739 | 0.002* | −0.137 | 0.911 |
| Q9*RT9 | 0.816 | 0.000** | 0.260 | 0.267 |
| Q10*RT10 | 0.621 | 0.014* | 0.667 | 0.012* |
| Q11*RT11 | −0.622 | 0.876 | −0.326 | 0.750 |
| Q12*RT12 | 0.619 | 0.005* | −0.312 | 0.818 |
| Q13*RT13 | 0.566 | 0.024* | −0.654 | 0.929 |
| Q14*RT14 | 0.229 | 0.223 | 0.153 | 0.367 |
| Q15*RT15 | 0.606 | 0.017* | −0.575 | 0.983 |
| Q16*RT16 | 0.267 | 0.241 | −0.620 | 0.835 |
| Q17*RT17 | 0.756 | 0.000** | 0.093 | 0.393 |
| Q18*RT18 | 0.556 | 0.034* | 0.002 | 0.497 |
| Q19*RT19 | 0.756 | 0.001* | 0.438 | 0.115 |
| Q20*RT20 | 0.814 | 0.000** | −0.027 | 0.562 |
| Q21*RT21 | 0.807 | 0.000** | 0.490 | 0.079 |

*p < 0.05; **p < 0.001.

(60.6%) had their asthma controlled in the previous month, as assessed by the asthma control test (ACT). Most asthmatics (49.0%) had mild intermittent asthma, while 5.0% of asthmatics had severe asthma. 99% of asthmatics participants were taking medication for asthma control.

## Confirmatory factor analysis

Factorial validity of questionnaire domains was evaluated using confirmatory factorial analysis (CFA). Fig 2 shows the structural model and factor loadings obtained, which were all statistically significant (p > 0.001) and equal to or greater than 0.30. The factorial structure is represented according to a model constituted by the four domains or factors to which the 21 items of the asthma self-knowledge questionnaire are related: General Aspects (Q1, Q2, Q3, Q4, Q5, Q6); Pathophysiological/ Clinical Aspects (Q7, Q8, Q9, Q10, Q11, Q12); Therapeutic Aspects (Q13, Q14, Q15, Q16, Q17, Q18); Non-pharmacological Therapeutic Aspects (Q19, Q20, Q21).

Regarding observed variables → latent factors (factor loadings): The path coefficients represent standardized factor loadings, indicating how strongly each item reflects its underlying factor. High loadings (≥0.7): Items such as Q4 (0.98 on Factor 1), Q15 (0.84 on Factor 3), and Q20 (0.88 on Factor 4) are strong indicators of their factors. Moderate loadings (0.4–0.7): Items like Q1 (0.41), Q3 (0.59), and Q10 (0.63) contribute meaningfully but less strongly. Lower loadings (<0.4): Items including Q2 (0.37), Q7 (0.31), Q9 (0.32), and Q16 (0.30) are modest indicators.

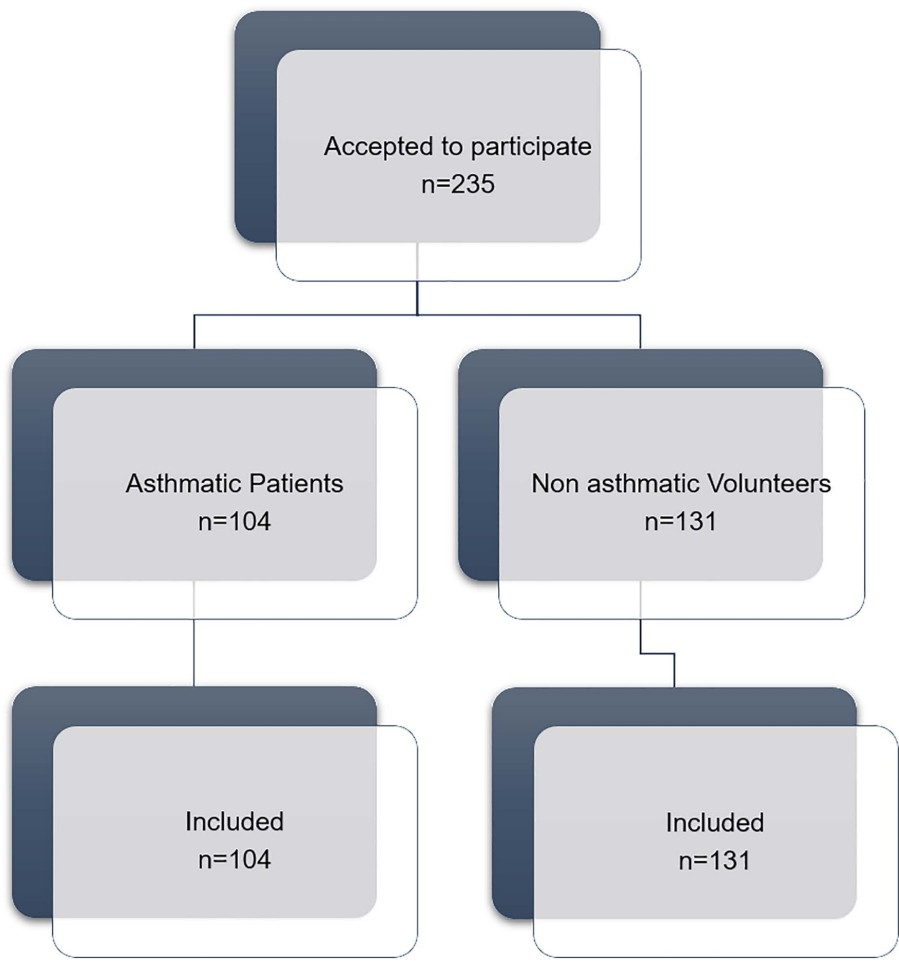

**Fig 1. Representative flowchart of the selection sample.**

Regarding latent factors ↔ latent factors (inter-factor correlations supporting construct validity): Factor 2 ↔ Factor 3 = 0.75, showing a strong relationship; Factor 2 ↔ Factor 4 = 0.73, indicating related constructs; Factor 3 ↔ Factor 4 = 0.46, a moderate relationship; and Factor 1 ↔ Factor 2 = 0.31, a modest relationship.

Regarding model implications: Some factors have items with excellent loadings (e.g., Factor 1 with Q4; Factor 4 with Q20 and Q21), making them strong measurement indicators. Overall, the model is reliable, with most items showing robust or acceptable coefficients, which boosts confidence in the validity of the factors.

The results of the CFA indicated that the four-factor structure provided a good fit to the data: $Chi^2 = 279.74$, $p < .001$; RMSEA = .043; GFI = .917; CFI = .898; PCLOSE = .07.

## Differences in self-knowledge about asthma between groups

As can be seen in Table 6, statistically significant differences were found in levels of asthma self-knowledge ($p < 0.001$), which indicates that asthmatic participants presented more knowledge (M = 3.87; SD = 0.43) when compared with participants without asthma (M = 3.62; SD = 0.42). Also, significant differences were found for all mean scores of the four domains of the self-knowledge questionnaire, confirming that asthmatic participants possess more asthma-related information.

**Table 4. Demographic features of the two sample groups.**

| Variable | Categories | All (n = 235) (n - %) | Asthmatics (n = 104) | Non-Asthmatics (n = 131) | t/ χ²; *p value* |
|---|---|---|---|---|---|
| Age (years) | Mean ± SD | 46.27 ± 15.85 | 44.26 ± 15.98 | 47.87 ± 15.62 | −1.714; p = 0.083 |
| | Range | 18–83 | 18 - 74 | 18–83 | |
| Age Group | ≤ 30 | 51 (21.7%) | 30 (28.8%) | 21 (16.0%) | 8.598; p = 0.072 |
| | 31–45 | 69 (29.4%) | 27 (26.0%) | 42 (32.1%) | |
| | 46–60 | 66 (28.1%) | 27 (26.0%) | 39 (29.8%) | |
| | ≥ 61 | 49 (20.8%) | 20 (19.2%) | 29 (22.2%) | |
| Gender | Female | 176 (74.9%) | 77 (74.0%) | 99 (75.6%) | 0.072; p = 0.788 |
| | Male | 59 (25.1%) | 27 (26.0%) | 32 (24.4%) | |
| Education | 1st cycle | 30 (12.8%) | 14 (13.5%) | 16 (12.2%) | 10.496; p = 0.062 |
| | 2nd cycle | 9 (3.8%) | 5 (4.8%) | 4 (3.1%) | |
| | 3rd cycle | 22 (9.4%) | 5 (4.8%) | 17 (13.0%) | |
| | High school | 59 (25.1%) | 21 (20.2%) | 38 (29.0%) | |
| | Professional | 31 (13.2%) | 19 (18.3%) | 12 (9.2%) | |
| | University | 84 (35.7%) | 40 (38.5%) | 44 (33.6%) | |
| Occupation | Unemployed | 27 (11.5%) | 10 (9.7%) | 13 (13.0%) | 2.126; p = 0.547 |
| | Employed | 152 (64.7%) | 67 (65.0%) | 85 (64.9%) | |
| | Retired | 40 (17.0%) | 17 (16.3%) | 23 (17.6%) | |
| | Student | 15 (6.4%) | 9 (8.7%) | 6 (4.6%) | |
| Residence | Urban | 139 (59.1%) | 66 (63.5%) | 73 (55.7%) | 4.765; p = 0.190 |
| | Village | 50 (21.3%) | 23 (22.1%) | 27 (20.6%) | |
| | Small village | 38 (16.2%) | 14 (13.5%) | 24 (18.3%) | |
| | Farm | 8 (3.4%) | 1 (1.0%) | 7 (5.3%) | |
| Asthma in family | Yes | 90 (38.4%) | 49 (47.1%) | 40 (31.3%) | 6.108; p = 0.013* |
| | No | 145 (61.6%) | 55 (52.9%) | 88 (68.8%) | |
| Contact with asthmatic people | Yes | 130 (55.2%) | 73 (70.2%) | 56 (43.0%) | 17.194; p = 0.000** |
| | No | 105 (44.3%) | 31 (29.8%) | 75 (55.7%) | |
| MMSE | Cognitive defect | 0 (0.0%) | 0 (0.0%) | 0 (0.0%) | 0.008; p = 0.927 |
| | No cognitive defect | 39 (16.6%) | 17 (16.3%) | 22 (16.8%) | |
| | Not applicable | 196 (83.4%) | 87 (83.7%) | 109 (83.2%) | |
| GDS | Normal | 20 (8.5%) | 9 (8.7%) | 11 (8.4%) | 4.699; p = 0.195 |
| | Light Depression | 15 (6.4%) | 9 (8.7%) | 6 (4.6%) | |
| | Severe Depression | 4 (1.7%) | 0 (0.0%) | 4 (3.1%) | |
| | Not applicable | 196 (83.4%) | 86 (82.7%) | 110 (84.0%) | |
| CES-D | Normal | 159 (68.0%) | 67 (64.4%) | 91 (69.2%) | 0.438; p = 0.803 |
| | Depressive | 38 (16.0%) | 18 (17.3%) | 19 (14.6%) | |
| | Not applicable | 38 (16.0%) | 16 (15.4%) | 21 (16.2%) | |

Note: t-student; χ² – Chi-square.

## Convergent validity

Lastly, we analyzed convergent validity by comparing the results of Asthma Self-Knowledge Questionnaire with those of the European Health Literacy Survey [1,23] designed to measure a similar construct. Results are shown in Table 7. Statistically significant associations (p < 0.001) were found between health literacy and global knowledge, general aspects, and non-pharmacological therapeutic aspects.

**Table 5. Clinical characterization of the asthmatic sample groups.**

| Years with asthma | 1–10 years | 46 (44.1%) |
|---|---|---|
| | 11–20 years | 27 (25.8%) |
| | 21–30 years | 16 (15.1%) |
| | 31–40 years | 12 (11.8%) |
| | 41–50 years | 3 (3.2%) |
| Asthma Control Test | Controlled | 63 (60.6%) |
| | Not controlled | 41 (39.4%) |
| Asthma Severity | Slightly intermittent | 50 (49.0%) |
| | Slightly persistent | 24 (24.0%) |
| | Moderately persistent | 22 (22.0%) |
| | Severely persistent | 4 (5.0%) |
| Asthma medication | Yes | 102 (99%) |
| | No | 1 (1.0%) |
| Other medication | Yes | 92 (89.2%) |
| | No | 12 (10.8%) |

## Discussion

To the best of our knowledge, this was the first study to develop and validate a self-knowledge questionnaire about bronchial asthma in Portugal, and one of few similar studies worldwide. In fact, we were able to develop and analyse face, content, and construct validity as well as reliability of the questionnaire to assess self-knowledge regarding asthma and demonstrate that it has acceptable psychometric properties and is simple to apply. Furthermore, it can detect significant differences between asthmatic and anon-asthmatic individuals (construct validity – known group). Although there are bronchial asthma self-knowledge questionnaires that have already been validated, in several countries, there is some heterogeneity across questionnaires [7–10], which has been reviewed by Pink et al [11]. Furthermore, several of these questionnaires are old and have outdated asthma concepts, and lack some additional aspects regarding asthma, which must be included to have adequate and complete tools to evaluate self-knowledge about the disease. Our questionnaire demonstrated good properties to be used in clinical settings.

For the development and validation of the questionnaire, we followed all validation criteria that are usually recommended [15–18]. Regarding the theoretical construct analysis, namely content validity, the results showed that only three of the twenty-four questions in the initial questionnaire had an I-CVI value <0.78, so the questions "The diagnosis of asthma can be confirmed by Chest X-ray", "Aspirin may trigger an asthma bout" and "The room of an asthmatic patient should always be closed" were removed from the final questionnaire, which consisted of 21 questions, and which is methodologically recommended [14].

As for the sensitivity analysis, data were considered as normally distributed, and reliability, assessed by internal consistency using the Cronbach's alpha coefficient produced an acceptable result. Also, reliability assessed using the test-retest presented results suggestive of good stability. The correlation coefficient (Spearman's Rho value) was used and indicated a significantly stronger correlation in the asthmatic group compared with the non-asthmatic group. This indicates that the questionnaire presented good group validity [16].

In relation to confirmatory factor analysis, we obtained values of Chi-square/degrees of freedom ratio of 2.129, which seems to be a good value because a value <3.00 is a good indicator. In the CFI index and the GFI very good values were not obtained, but the PCFI and the PGFI indicated an acceptable model fit, as well as the RMSEA. Although we only included 235 individuals, this number is within an adequate sample size for this type of analysis, since the subjects/ items ratio was 11.19, which is higher than the minimum threshold of a ratio of 3, and the number of recruited individuals was well above the recommended minimum [31].

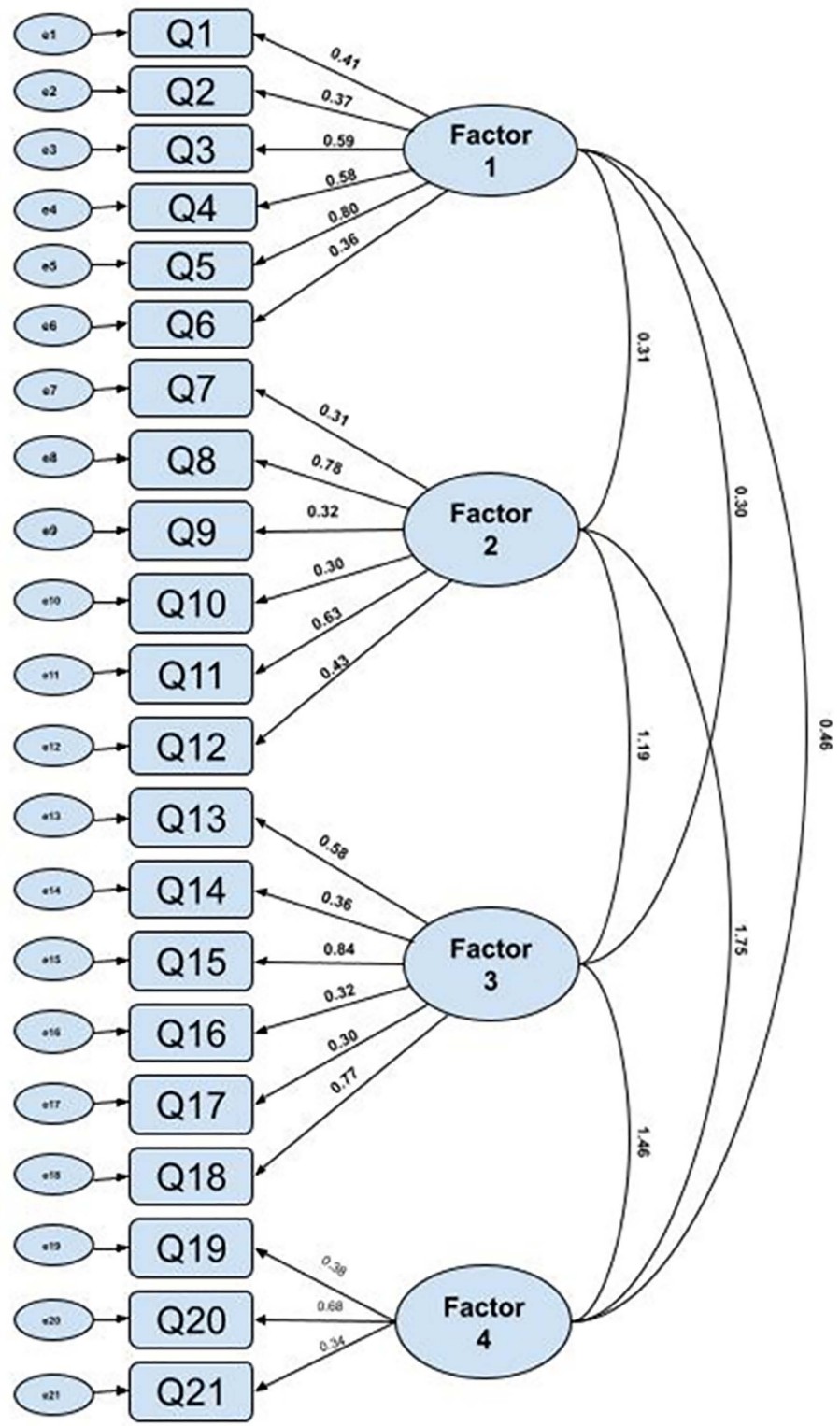

**Fig 2. Structural model.**

**Table 6. Mean Scores obtained in the Asthma Self-Knowledge Questionnaire by groups.**

| Variable | Category | n | Mean±SD | p value |
|---|---|---|---|---|
| Global Knowledge | Asthmatic | 104 | 3.87±0.43 | 0.000** |
| | Non-asthmatic | 131 | 3.62±0.42 | |
| General Aspects | Asthmatic | 104 | 4.30±0.50 | 0.049* |
| | Non-asthmatic | 131 | 4.15±0.58 | |
| Pathophysiological/ Clinical Aspects | Asthmatic | 104 | 3.59±0.47 | 0.000** |
| | Non-asthmatic | 131 | 3.32±0.61 | |
| Therapeutic Aspects | Asthmatic | 104 | 3.65±0.69 | 0.000** |
| | Non-asthmatic | 131 | 3.32±0.64 | |
| Non-pharmacological Therapeutic Aspects | Asthmatic | 104 | 4.00±0.76 | 0.025* |
| | Non-asthmatic | 131 | 3.78±0.69 | |

**Table 7. Correlation matrix.**

| | 1 | 2 | 3 | 4 | 5 | 6 |
|---|---|---|---|---|---|---|
| 1-Global Knowledge | 1 | | | | | |
| 2-General Aspects | 0.645** | 1 | | | | |
| 3-Pathophysiological/ Clinical Aspects | 0.691** | 0.349** | 1 | | | |
| 4-Therapeutic Aspects | 0.754** | 0.198** | 0.276** | 1 | | |
| 5-Non-pharmacological Therapeutic Aspects | 0.631** | 0.215** | 0.224** | 0.483** | 1 | |
| 6-Health Literacy | 0.147* | .138* | 0.016 | 0.124 | 0.136* | 1 |

Statistically significant differences in levels of asthma self-knowledge were obtained between groups, i.e., in the asthmatic group very high knowledge levels were obtained while in the non-asthmatic group high knowledge levels were obtained. This was one of the hypotheses that we wanted to test, so we can conclude that having bronchial asthma is associated with greater knowledge about the disease.

Another hypothesis that we wanted to test was whether there would be differences in self-knowledge about specific issues of asthma between groups and, in fact, statistically significant differences were observed, namely in terms of the Pathophysiological/ Clinical Aspects and in Therapeutic Aspects domains. Both in the asthmatic group and in the non-asthmatic group, there was greater knowledge regarding the General Aspects and he Nonpharmacological Therapeutic Aspects, but a lower degree of knowledge regarding the Pathophysiological/ Clinical Aspects. Another study also on validation of an asthma self-knowledge questionnaire also divided their questions into several domains; however, we cannot compare our results with those from that study since the authors did not analyze differences of knowledge within each domain [10].

Regarding factors that might affect self-knowledge scores, there was a significant correlation between health literacy and knowledge scores, which may hint at health literacy significantly affecting asthma self-knowledge. This is in line with the study of Mancuso & Rincon [6], where the authors demonstrated that health literacy significantly influenced knowledge of asthma and self-management.

Our study has several limitations. This was a convenience sample selected by non-probabilistic methods, which impedes generalization of results. Secondly, social desirability and the voluntary effect may have played an influence, and this may have biased responses to the questionnaire. Since the questionnaires were mainly collected at hospital centers, there may have been a bias in the sample, in the sense that people seeking health care are likely to be more interested and more informed, particularly those asthmatic patients regularly seen at specialty clinics, and which may not

be representative of most asthmatics. Finally, the bronchial asthma self-knowledge questionnaire, as well as the MMSE, GDS, CES-D, and EHLS are based upon self-report, which is subject to being influenced by humor-related or memory biases. Overall, we endeavored to keep bias in self-reported items to a minimum by ensuring anonymity to promote honest responses, and also by using attempting to reduce social desirability bias by informing that patients were not being assessed by their "right" or "wrong" answers. Furthermore, we also believe that information bias was reduced by running a pilot study, using objective preexisting international criteria, and keeping participants blinded to the hypotheses under investigation. In any case, future studies should increase the awareness of the possible shortcomings and pitfalls of decision making that can result in various types of bias associated with self-report.

## Conclusions

In conclusion, we developed a short questionnaire concerning asthma knowledge, which showed adequate psychometric properties. However, some aspects still must be optimized. Using this questionnaire, we have confirmed that having bronchial asthma is associated with greater knowledge about the disease. Finally, in this study, health literacy only marginally affects asthma self-knowledge but such findings need to be thoroughly checked using univariate and multivariate analyses. Future studies with a larger sample sizes and improved study strategy are required to clarify aspects related to factors that influence knowledge about bronchial asthma.

## Supporting information

**S1 Table. Asthma Questionnaire (English).**
(PDF)

## Acknowledgments

We would like to acknowledge all immunoallergologists who contributed to calculating the CVI-I: Ana Morete (Allergy Unit, Instituto and Hospital CUF, Porto, Portugal), Ana Todo-Bom (Immunoallergology Service, Coimbra University Hospital Centre, Coimbra, Portugal), André Moreira (Immunoallergology Service, São João University Hospital Center, Porto, Portugal), Carlos Nunes (Centro de Imunoalergologia do Algarve, Portimão, Portugal), Carlos Lozoya-Ibáñez (Allergy, Hospital Amato Lusitano, Unidade Local de Saúde, Castelo Branco, Portugal), Emília Faria (Immunoallergology Service, Coimbra University Hospital Centre, Coimbra, Portugal), Elsa Tomaz (Immunoallergology Department, Centro Hospitalar de Setúbal, Setúbal, Portugal), Filipe Inácio (Immunoallergology Department, Centro Hospitalar de Setúbal, Setúbal, Portugal), Helena Falcão (Immunoallergology Service, Centro Hospitalar Universitário do Porto, Porto, Portugal), Joana Gomes Belo (Immunoallergology Department, Cova da Beira Hospital Center, Covilhã, Portugal), João Almeida Fonseca (CINTESIS@RISE, MEDCIDS, Faculty of Medicine of the University of Porto, Porto, Portugal), José A. Ferreira (Immunoallergology Service, Centro Hospitalar Vila Nova de Gaia/Espinho, Vila Nova de Gaia, Portugal), Leonor Leão (Immunoallergology Service, São João University Hospital Center, Porto, Portugal), Manuel Branco Ferreira (Immunoallergology Department, Centro Hospitalar Lisboa Norte, Lisbon, Portugal), Paula Alendouro (Immunoallergology Service, Centro Hospitalar do Alto Ave, Guimarães/ Fafe, Portugal), Pedro Mata (ICA – Instituto Clínico de Alergologia, Lisbon, Portugal).

## Author contributions

**Conceptualization:** Henrique Pereira, Ana Rita Pedro, Luís Taborda-Barata.

**Data curation:** Henrique Pereira, Luís Taborda-Barata.

**Formal analysis:** Adalberto Fernandes Dos Santos, Henrique Pereira.

**Investigation:** Adalberto Fernandes Dos Santos, Renata Costa.

**Methodology:** Henrique Pereira.

**Project administration:** Luís Taborda-Barata.

**Supervision:** Luís Taborda-Barata.

**Validation:** Renata Costa, Henrique Pereira.

**Visualization:** Henrique Pereira.

**Writing – original draft:** Adalberto Fernandes Dos Santos, Renata Costa, Henrique Pereira.

**Writing – review & editing:** Adalberto Fernandes Dos Santos, Renata Costa, Henrique Pereira, Ana Rita Pedro, Luís Taborda-Barata.

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
