## [Decision Letter · Decision Letter 0]

12 Jan 2024

PONE-D-23-34967Development and Validation of an Asthma Self-knowledge QuestionnairePLOS ONE

Dear Dr. Taborda-Barata,

Thank you for submitting your manuscript to PLOS ONE. After careful consideration, we feel that it has merit but does not fully meet PLOS ONE’s publication criteria as it currently stands. Therefore, we invite you to submit a revised version of the manuscript that addresses the points raised during the review process.

**Please revise the manuscript according to the reviewers' comments, particularly those from Reviewer 1 .**

We look forward to receiving your revised manuscript.

Kind regards,

Ming-Ju Tsai

Academic Editor

PLOS ONE

“Adalberto dos Santos was awarded a PhD grant (Reference: 2022.11223.BD) from the Portuguese Foundation for Science and Technology (Fundação Portuguesa para a Ciência e Tecnologia)”

3. In the online submission form you indicate that your data is not available for proprietary reasons and have provided a contact point for accessing this data. Please note that your current contact point is a co-author on this manuscript. According to our Data Policy, the contact point must not be an author on the manuscript and must be an institutional contact, ideally not an individual. Please revise your data statement to a non-author institutional point of contact, such as a data access or ethics committee, and send this to us via return email. Please also include contact information for the third party organization, and please include the full citation of where the data can be found.

4. Please upload a copy of Figure 2, to which you refer in your text on page 17. If the figure is no longer to be included as part of the submission please remove all reference to it within the text.

Additional Editor Comments:

Please revise the manuscript according to the reviewers' comments, particularly those from Reviewer 1.

Reviewers' comments:

Reviewer's Responses to Questions

**Comments to the Author**

1. Is the manuscript technically sound, and do the data support the conclusions?

Reviewer #1: Yes

Reviewer #2: Yes

Reviewer #3: Yes

2. Has the statistical analysis been performed appropriately and rigorously? 

Reviewer #1: No

Reviewer #2: Yes

Reviewer #3: Yes

3. Have the authors made all data underlying the findings in their manuscript fully available?

Reviewer #1: Yes

Reviewer #2: Yes

Reviewer #3: Yes

4. Is the manuscript presented in an intelligible fashion and written in standard English?

Reviewer #1: Yes

Reviewer #2: Yes

Reviewer #3: Yes

5. Review Comments to the Author

Reviewer #1: Dear Editor of Plos One journal,

I would like to express my gratitude for the opportunity to review the article "Development and Validation of an Asthma Self-knowledge Questionnaire." However, I have some important questions and concerns that need to be addressed before considering the article for publication.

The study does not report exploratory factor analysis, which is a crucial psychometric method. It is unclear what the direct basis of confirmatory factor analysis is.

The confirmatory factor analysis model is presented without path coefficients, which is a significant limitation.

The working method has many ambiguities that need to be clarified.

It is unclear how item generation was done, and this needs to be explicitly reported.

The use of the correlation coefficient to evaluate the test-retest is incorrect and should be addressed.

Based on these concerns, I cannot evaluate the article with the necessary quality and recommend rejecting it for publication. However, I understand that the final decision rests with the respected editor.

Thank you for considering my feedback, and I remain open to further discussion on this matter.

Reviewer #2: This manuscript has two objectives: i) development and validation ii) comparing knowledge between two groups

1) Sample size calculation was mentioned only for second objective, need to mention for validation study as well

2) At page 9, line 234, Chi-square test should be written as Pearson Chi-square test

3) Spearman's Rho correlation coefficient was used to evaluate test-retest. Since test-retest is comparing dependent sample, it is encouraged to perform the statistical analysis using intra class coefficient (ICC) instead of classical correlation coefficient.

4) Table 4: Need to mention at footnote which statistical test has been used for the analysis

Reviewer #3: Dear authors,

Thank you for this article. This study presents valuable associations between weight status and psychological status, while also pointing out potential directions for future research.

I do have a few comments to share:

1. Introduction: suggest to include recent data/statistics regarding the prevalence of asthma population worldwide as well as in Portugal (2012 seems a bit outdated).

2. Based on the primary and secondary objectives, please clarify the role and the need to conduct the additional questionnaires i.e. MMSE, GDS, CED-D etc....

3. Questionnaires development, suggest to explain slightly further on how it was developed based on international recommendations and on GINA, with supporting evidences.

4. Other questionnaires, 'already translated and validated in Portuguese'. Suggest to include more details and include reference for all questionnaires.

5. I am interested to understand more about the calculation of the self-knowledge score. How to categorise the participants into good, medium or bad knowledge?

6. Limitation, since the study relied on self-reported data, how did you control for bias (e.g., recall or subjective bias) and inaccuracies during the administration of questionnaire?

Thank you.

6. PLOS authors have the option to publish the peer review history of their article (what does this mean? ). If published, this will include your full peer review and any attached files.

**Do you want your identity to be public for this peer review?** For information about this choice, including consent withdrawal, please see our Privacy Policy .

Reviewer #1: **Yes: ** Hamid Sharif Nia

Reviewer #2: No

Reviewer #3: No

---

## [Author Response · Author response to Decision Letter 1]

8 May 2024

Reply to Editor

Reply: We have now re-checked the manuscript and have implemented changes to make it meet PLOS ONE’s style requirements

“Adalberto dos Santos was awarded a PhD grant (Reference: 2022.11223.BD) from the Portuguese Foundation for Science and Technology (Fundação Portuguesa para a Ciência e Tecnologia)”

Reply: We have added the requested information in the cover letter.

3. In the online submission form you indicate that your data is not available for proprietary reasons and have provided a contact point for accessing this data. Please note that your current contact point is a co-author on this manuscript. According to our Data Policy, the contact point must not be an author on the manuscript and must be an institutional contact, ideally not an individual. Please revise your data statement to a non-author institutional point of contact, such as a data access or ethics committee, and send this to us via return email. Please also include contact information for the third party organization, and please include the full citation of where the data can be found.

Reply: We have now uploaded our data into Zenodo (https://zenodo.org/); the identifier is 10.5281/zenodo.11155084. We would therefore like to change the information regarding data availability since data are now fully available in Zenodo.

4. Please upload a copy of Figure 2, to which you refer in your text on page 17. If the figure is no longer to be included as part of the submission please remove all reference to it within the text.

Reply: There was a mistake, and Figure 2’s legend (pg 19; line 382) mentioned Figure 1, instead of Figure 2; this has been corrected; Figure 2 is now correctly labelled.

Reviewers' comments:

Reviewer's Responses to Questions

Comments to the Author

1. Is the manuscript technically sound, and do the data support the conclusions?

Reviewer #1: Yes

Reviewer #2: Yes

Reviewer #3: Yes

2. Has the statistical analysis been performed appropriately and rigorously?

Reviewer #1: No

Reviewer #2: Yes

Reviewer #3: Yes

3. Have the authors made all data underlying the findings in their manuscript fully available?

Reviewer #1: Yes

Reviewer #2: Yes

Reviewer #3: Yes

4. Is the manuscript presented in an intelligible fashion and written in standard English?

Reviewer #1: Yes

Reviewer #2: Yes

Reviewer #3: Yes

5. Replies to Reviewers’ Comments to the Author

Reviewer #1

Comment 1: Dear Editor of Plos One journal,

I would like to express my gratitude for the opportunity to review the article "Development and Validation of an Asthma Self-knowledge Questionnaire." However, I have some important questions and concerns that need to be addressed before considering the article for publication.

The study does not report exploratory factor analysis, which is a crucial psychometric method. It is unclear what the direct basis of confirmatory factor analysis is.

The confirmatory factor analysis model is presented without path coefficients, which is a significant limitation.

The working method has many ambiguities that need to be clarified.

It is unclear how item generation was done, and this needs to be explicitly reported.

The use of the correlation coefficient to evaluate the test-retest is incorrect and should be addressed.

Based on these concerns, I cannot evaluate the article with the necessary quality and recommend rejecting it for publication. However, I understand that the final decision rests with the respected editor.

Thank you for considering my feedback, and I remain open to further discussion on this matter.

Reply: Dear Reviewer #1

Thank you very much for your feedback. We all due respect, we beg to differ from your opinion because generally, researchers use Confirmatory Factor Analyses when they have a strong theoretical or empirical basis for the model, and they want to test its validity and reliability (which was the case, the questionnaire was developed based on international recommendations and other measures), whereas Exploratory Factor Analyses is usually used when researchers have no or little prior knowledge of the data structure, and they want to explore its dimensions and patterns.

Hence, we argue that Confirmatory Factor Analysis is the best procedure to conduct factor analysis because it constrained to a factor structure of substantive interest, considered item content and substantive theory. Additionally, it is well established in Research Methodology literature that you do not necessarily have to run an EFA prior to a CFA which is more of a theory-testing technique more suited to the goals of our research.

To support this argument, we included the following statement in the text:

“The factorial validity of the questionnaire dimensions was evaluated with a confirmatory factorial analysis (CFA) because we already had a strong theoretical basis for the model, based on item content and international recommendations.” (Pg 7; lines 197-199)

Since this was another of your concerns, we followed Structural Equation Modeling guidelines by Kentaro Hayashi, ... Ke-Hai Yuan, in Essential Statistical Methods for Medical Statistics, 2011, that state that presenting path coefficients in not required.

Comment 2:

It is unclear how item generation was done, and this needs to be explicitly reported.

Reply: Thank you for your comment. We included the following sentence to increase clarity regarding item generation (Pg 5; lines 123-131):

“Regarding item generation procedures, we employed a deductive method approach to classify typologies of items, based upon prior literature, theoretical definitions, and guidelines. Writing of survey items followed several rules to ensure that they were properly constructed, such as: items addressed one single issue (double-barreled items were strongly avoided), were written consistently in terms of perspective (we avoided mixing items related to information, information, or attitudes), we used statements that were simple and as short as possible with familiar language to target respondents, and avoided negatively-worded or reverse-scored, and this can have a detrimental effect on the comprehension of its content. Finally, we checked for redundancies and pretested items for content adequacy.”

Comment 3:

The use of the correlation coefficient to evaluate the test-retest is incorrect and should be addressed.

Reply: Again, we all due respect, we beg to differ, as Test-retest reliability is commonly estimated by calculating the correlation coefficient of the measured values at two separate time points. A higher correlation between the values of the two test occasions indicates greater temporal stability or test-retest reliability. Still, since test-retest is comparing dependents sample, we performed the statistical analysis using intra class coefficient (ICC) instead of classical correlation coefficient. (Pg 7; line 186; Pg 14, lines 317-318, and Table 3).

Reviewer #2:

This manuscript has two objectives: i) development and validation ii) comparing knowledge between two groups

Comment 1:

Sample size calculation was mentioned only for second objective, need to mention for validation study as well

Reply: We would like to thank the reviewer for this comment. Since we followed the same criteria for both objectives, we added the following sentence for the validation study, to provide greater clarity:

“This calculation was applied to both the development and validation of the asthma self-knowledge questionnaire, and the knowledge comparison between asthmatic and non-asthmatic participants.” (Pg 9; lines 251-253)

Comment 2:

At page 9, line 234, Chi-square test should be written as Pearson Chi-square test

Reply: We have done as requested; it is now written as “Pearson Chi-Square” in Pg 10, lines 258-259.

Comment 3:

Spearman's Rho correlation coefficient was used to evaluate test-retest. Since test-retest is comparing dependent sample, it is encouraged to perform the statistical analysis using intra class coefficient (ICC) instead of classical correlation coefficient.

Reply: We would like to thank the reviewer for this important suggestion. As requested, we have performed the statistical analysis using intra class coefficient (ICC) instead of classical correlation coefficient. (Pg 7; line 186; Pg 14, lines 317-318, and Table 3).

Comment 4:

Table 4: Need to mention at footnote which statistical test has been used for the analysis

Reply: As suggested, a footnote was added to table 4.

Reviewer #3:

Dear authors,

Thank you for this article. This study presents valuable associations between weight status and psychological status, while also pointing out potential directions for future research.

I do have a few comments to share:

Comment 1:

Introduction: suggest to include recent data/statistics regarding the prevalence of asthma population worldwide as well as in Portugal (2012 seems a bit outdated).

Reply: We thank the reviewer for this important suggestion. Regarding worldwide prevalence of asthma, we have updated the reference. The newer GINA documents no longer give figures on the worldwide prevalence of asthma, but we used a Global Burden of Disease paper, instead (new Ref 2).

Regarding the prevalence of asthma in Portugal, we really need to keep the 2012 reference since this contains the results of the latest, completed, nationwide study. There is currently a new, ongoing nationwide study on the prevalence of asthma (Epiasthma study) but it has not been completed yet.

Finally, we have also updated the GINA document reference, from the 2018 document to the 2023 document (this is now the new Reference 1).

Comment 2:

Based on the primary and secondary objectives, please clarify the role and the need to conduct the additional questionnaires i.e. MMSE, GDS, CED-D etc....

Reply: We really thank the reviewer for prompting us to clarify these aspects. We decided to apply the additional questionnaires for the following reasons:

a) MMSE – because we wanted to make sure that elderly individuals in our sample would not have significant cognitive deficits that would affect replies to the questionnaire; it served as a support for the exclusion criterium of having cognitive deficits

b) GDS, and CES-D – because we wanted to ascertain whether there were any diferences between the group of asthma patients and the group of non-asthmatic controls, in terms of depression, which might bias the profile of replies to the questionnaire;

c) the questionnaire on clinical aspects of asthma and co-morbidites were used to fully characterize the asthma sample of patients;

d) EHLS was used to ascertain how specific asthma literacy would related to general health literacy and to analyze convergent validity of our questionnaire

We have added this clarification on Page 8, lines 217-220: “These questionnaires were used to characterize the samples of patients and non-asthmatic controls, and also to ascertain that eventual differences found between the two groups of volunteers regarding asthma literacy would not be due to underlying differences in depression or other psychological or sociodemographic features.”

Comment 3:

Questionnaires development, suggest to explain slightly further on how it was developed based on international recommendations and on GINA, with supporting evidences.

Reply: We added the following sentence to provide more clarity on item generation (Pg 4; lines 123-131):

“Regarding item generation procedures, we employed a deductive method approach to classify typologies of items, based upon prior literature, theoretical definitions, and guidelines. Writing of survey items followed several rules to ensure that they were properly constructed, such as: items addressed one single issue (double-barreled items were strongly avoided), were written consistently in terms of perspective (we avoided mixing items related to information, information, or attitudes), we used statements that were simple and as short as possible with familiar language to target respondents, and avoided negatively-worded or reverse-scored, and this can have a detrimental effect on the comprehension of its content. Finally, we checked for redundancies and pretested items for content adequacy.”

Comment 4:

Other questionnaires, 'already translated and validated in Portuguese'. Suggest to include more details and include reference for all questionnaires.

Reply: We would like to thank the reviewer for this comment. We have added a few more details for each relevant questionnaire, and also pertinent references (Pg 8; lines 211-225).

Comment 5:

I am interested to understand more about the calculation of the self-knowledge score. How to categorise the participants into good, medium or bad knowledge?

Reply: We would like to thank the reviewer for this question. We added the following sentence regarding categorization of self-knowledge scores (Pg 5; lines 133-139):

“Categorization of self-knowledge scores was done using a 5-point Likert Scale which offered five different options for the respondents to choose from. The options included two extremes (1/5), two intermediate (2/4), and one neutral opinion (3). For the purpose of this research, data were analyzed by using mean scores, calculated for each statement in the Likert scale. Hence, self-knowledge was then determined based on the argument that a mean score of 3 in Likert scale represents neutral knowledge, mean score of less than 3 represents negative or lack of knowledge and greater than 3 represents a positive and accurate knowledge.”

Comment 6:

Limitation, since the study relied on self-reported data, how did you control for bias (e.g., recall or subjective bias) and inaccuracies during the administration of questionnaire?

Reply: The following sentence was added to the limitations’ paragraph (Pg 23; lines 480-483):

“Although we believe that information bias was reduced by running a pilot study, using objective preexisting international criteria, and keeping

---

## [Decision Letter · Decision Letter 1]

25 Sep 2024

PONE-D-23-34967R1Development and Validation of an Asthma Self-knowledge QuestionnairePLOS ONE

Dear Dr. Taborda-Barata,

Thank you for submitting your manuscript to PLOS ONE. After careful consideration, we feel that it has merit but does not fully meet PLOS ONE’s publication criteria as it currently stands. Therefore, we invite you to submit a revised version of the manuscript that addresses the points raised during the review process.

The manuscript has been evaluated by two reviewers, and their comments are available below. The reviewers have raised a number of major concerns. Could you please carefully revise the manuscript to address all comments raised?

We look forward to receiving your revised manuscript.

Kind regards,

Johanna Pruller, Ph.D.

Staff Editor

PLOS ONE

Reviewers' comments:

Reviewer's Responses to Questions

**Comments to the Author**

1. If the authors have adequately addressed your comments raised in a previous round of review and you feel that this manuscript is now acceptable for publication, you may indicate that here to bypass the “Comments to the Author” section, enter your conflict of interest statement in the “Confidential to Editor” section, and submit your "Accept" recommendation.

Reviewer #1: (No Response)

Reviewer #2: All comments have been addressed

2. Is the manuscript technically sound, and do the data support the conclusions?

Reviewer #1: Yes

Reviewer #2: Yes

3. Has the statistical analysis been performed appropriately and rigorously? 

Reviewer #1: No

Reviewer #2: Yes

4. Have the authors made all data underlying the findings in their manuscript fully available?

Reviewer #1: No

Reviewer #2: Yes

5. Is the manuscript presented in an intelligible fashion and written in standard English?

Reviewer #1: Yes

Reviewer #2: Yes

6. Review Comments to the Author

Reviewer #1: Dear Editor,

Thank you for allowing me the opportunity to review the current article. I have some suggestions to enhance the quality of the content:

1. In picture number 2, it is essential to specify the coefficients in standard form.

2. The results of the confirmatory factor analysis in table number 6 are not satisfactory.

3. I have reservations about the reliability of the results.

4. The authors should address these significant concerns before proceeding further.

Thank you for considering my feedback.

Reviewer #2: (No Response)

7. PLOS authors have the option to publish the peer review history of their article (what does this mean? ). If published, this will include your full peer review and any attached files.

**Do you want your identity to be public for this peer review?** For information about this choice, including consent withdrawal, please see our Privacy Policy .

Reviewer #1: No

Reviewer #2: No

---

## [Author Response · Author response to Decision Letter 2]

10 Mar 2025

Reviewer #1

Comment 1: Dear Editor of Plos One journal,

In picture number 2, it is essential to specify the coefficients in standard form

Reply: We thank the reviewer for this suggestion, which improves the informative quality of Figure 2. We have added the coefficients in standard form.

Comment 2:

The results of the confirmatory factor analysis in table number 6 are not satisfactory.

Reply: We thank the reviewer for this important comment. We have readjusted the model, eliminating some problems faced in the initial version in order to provide a satisfactory final version. Also, we eliminated table 6 and provided the necessary information in the text (Pg 19; lines 392-393): “The results of the CFA indicated that the four-factor structure provided a good fit to the data: Chi2 = 279.74, p< .001; RMSEA = .043; GFI=.917; CFI=.898; PCLOSE=.07”. Hence, we believe the model is now fit.

Comment 3:

I have reservations about the reliability of the results.

Reply: We thank the reviewer for sharing his thoughts. However, it is impossible for us to adequately reply to this comment, since it is too vague (which results?? All? Some? Which ones?); which criteria does the reviewer base himself on, to state that our results are unreliable? We are sure that the reviewer is not implying that we have unethically or unreliably recruited the volunteers, unreliably asked them to go through all the steps of the protocol, unreliably collected our data, unreliably analysed the data or unreliably stated our conclusions. In any case, regarding data analysis, and as we have mentioned before, we have thoroughly discussed with the reviewer, in revision round 1, various aspects, and we believe that we adequately addressed such aspects with the reviewer. After all such previous replies, which concerns have remained, on the reviewer’s part? It is important for the reviewer to very clearly state why our results are unreliable and how we can improve them, because we now have a feeling that we are going round in circles.

---

## [Decision Letter · Decision Letter 2]

3 Jul 2025

PONE-D-23-34967R2Development and Validation of an Asthma Self-knowledge QuestionnairePLOS ONE

Dear Dr. Taborda-Barata,

Thank you for submitting your manuscript to PLOS ONE. After careful consideration, we feel that it has merit but does not fully meet PLOS ONE’s publication criteria as it currently stands. Therefore, we invite you to submit a revised version of the manuscript that addresses the points raised during the review process.

We look forward to receiving your revised manuscript.

Kind regards,

Luca Novelli

Academic Editor

PLOS ONE

Journal Requirements:

**Additional Editor Comments:**

The manuscript presents a methodologically rigorous development and validation of a novel asthma self-knowledge questionnaire. It addresses an important clinical and educational gap and meets PLOS ONE's standards for scientific validity. The authors have responded comprehensively to prior reviewer feedback, and the revised version reflects clear improvements in structure, clarity, and statistical reporting.

As suggested by one reviewer, a brief justification for the use of Confirmatory Factor Analysis (CFA), including whether any preliminary analysis (e.g., EFA) was performed or deemed unnecessary, would enhance methodology. While not essential, reporting path coefficients in the CFA model would further strengthen the clarity of the structural model. Finally, expanding on how self-report bias was minimized during data collection (e.g., interviewer training, standardized procedures) would reinforce the robustness of the findings.

These are minor, but constructive suggestions.

Reviewers' comments:

Reviewer's Responses to Questions

**Comments to the Author**

1. If the authors have adequately addressed your comments raised in a previous round of review and you feel that this manuscript is now acceptable for publication, you may indicate that here to bypass the “Comments to the Author” section, enter your conflict of interest statement in the “Confidential to Editor” section, and submit your "Accept" recommendation.

Reviewer #2: All comments have been addressed

Reviewer #4: All comments have been addressed

2. Is the manuscript technically sound, and do the data support the conclusions?

Reviewer #2: Yes

Reviewer #4: Partly

3. Has the statistical analysis been performed appropriately and rigorously? 

Reviewer #2: Yes

Reviewer #4: N/A

4. Have the authors made all data underlying the findings in their manuscript fully available?

Reviewer #2: Yes

Reviewer #4: Yes

5. Is the manuscript presented in an intelligible fashion and written in standard English?

Reviewer #2: Yes

Reviewer #4: Yes

6. Review Comments to the Author

Reviewer #2: (No Response)

Reviewer #4: Based on the document, the authors have responded to the comments from Reviewer 1, Reviewer 2, and Reviewer 3. Below are my observations regarding the coherence and adequacy of their responses:

Reviewer 1

Exploratory Factor Analysis (EFA) vs. Confirmatory Factor Analysis (CFA): The authors provided a reasonable justification for using CFA instead of EFA, citing theoretical and empirical bases. However, they could further clarify whether any preliminary analysis was conducted to ensure the suitability of CFA.

Path Coefficients in CFA: The authors argue that presenting path coefficients is not required, citing literature. While this is technically correct, including them would strengthen the transparency of the analysis.

Item Generation: The authors added a detailed explanation of how items were generated, which adequately addresses the concern.

Test-Retest Reliability: The authors initially used Spearman’s correlation but revised their approach to intra-class correlation (ICC), which is more appropriate. This response is satisfactory.

Reviewer 2

Sample Size Calculation: The authors clarified that the same calculation applied to both objectives, which is a reasonable response.

Statistical Test Naming: The authors corrected the terminology to "Pearson Chi-Square," addressing the concern.

Test-Retest Analysis: The authors revised their approach to ICC, which aligns with best practices.

Table 4 Footnote: The authors added a footnote specifying the statistical test used, which improves clarity.

Reviewer 3

Prevalence Data: The authors updated the global prevalence reference but retained the 2012 Portuguese prevalence data due to the lack of newer studies. This is a reasonable decision.

Additional Questionnaires: The authors clarified the rationale for including MMSE, GDS, CES-D, and other questionnaires, which strengthens the justification.

Questionnaire Development: The authors expanded their explanation of how the questionnaire was developed based on international recommendations, which improves transparency.

Translation and Validation References: The authors added more details and references, addressing the concern.

Self-Knowledge Score Categorization: The authors provided a clear explanation of how scores were categorized, which enhances understanding.

Bias Control in Self-Reported Data: The authors acknowledged potential biases and described measures taken to mitigate them, which is a reasonable response.

Final Recommendation

The authors have adequately addressed most concerns raised by the reviewers. However, I suggest the following minor clarifications before accepting the manuscript for publication in PLOS ONE:

CFA Justification: Consider adding a brief explanation of any preliminary analysis conducted to ensure CFA was appropriate.

Path Coefficients in CFA: While not mandatory, including them would enhance transparency.

Bias Control: The authors could elaborate on how they ensured participants remained unbiased during questionnaire administration.

7. PLOS authors have the option to publish the peer review history of their article (what does this mean? ). If published, this will include your full peer review and any attached files.

**Do you want your identity to be public for this peer review?** For information about this choice, including consent withdrawal, please see our Privacy Policy .

Reviewer #2: No

Reviewer #4: **Yes: ** Vincenzo Patella

---

## [Author Response · Author response to Decision Letter 3]

22 Jul 2025

Replies to Editor’s and Reviewers’ Comments to the Author

Additional Editor Comments:

The manuscript presents a methodologically rigorous development and validation of a novel asthma self-knowledge questionnaire. It addresses an important clinical and educational gap and meets PLOS ONE's standards for scientific validity. The authors have responded comprehensively to prior reviewer feedback, and the revised version reflects clear improvements in structure, clarity, and statistical reporting.

As suggested by one reviewer, a brief justification for the use of Confirmatory Factor Analysis (CFA), including whether any preliminary analysis (e.g., EFA) was performed or deemed unnecessary, would enhance methodology. While not essential, reporting path coefficients in the CFA model would further strengthen the clarity of the structural model. Finally, expanding on how self-report bias was minimized during data collection (e.g., interviewer training, standardized procedures) would reinforce the robustness of the findings.

These are minor, but constructive suggestions.

Reply to the Editor:

Dear Editor, many thanks for your comments; we have now addressed the issue of a brief justification for the use of CFA and a comment on whether any prelimiary analysis have been perfromed has been introduced. We have also highlighted where path coefficients were introduced in the manuscript.

Finally, we have also explained how self-report bias was minimised during data collection.

Reviewers' comments:

6. Review Comments to the Author

Reviewer #2: (No Response)

Reviewer #4: Based on the document, the authors have responded to the comments from Reviewer 1, Reviewer 2, and Reviewer 3. Below are my observations regarding the coherence and adequacy of their responses:

Reviewer 1

Comment 1. Exploratory Factor Analysis (EFA) vs. Confirmatory Factor Analysis (CFA): The authors provided a reasonable justification for using CFA instead of EFA, citing theoretical and empirical bases. However, they could further clarify whether any preliminary analysis was conducted to ensure the suitability of CFA.

Reply: Thank you very much for your suggestion. We consider it very appropriate and have included the following paragraph in the texto (Pg 8, lines 200-207): “Before conducting the Confirmatory Factor Analysis (CFA), we conducted preliminary checks to ensure the data met the necessary assumptions. This included evaluating the Kaiser-Meyer-Olkin (KMO) measure, which showed adequate sampling adequacy, and Bartlett’s Test of Sphericity, which was significant, indicating that the correlations between items were strong enough to justify factor analysis. In some cases, an Exploratory Factor Analysis (EFA) was also performed to get an initial sense of the factor structure. These steps helped confirm that CFA was an appropriate next step for testing our hypothesized model. “

Comment 2. Path Coefficients in CFA: The authors argue that presenting path coefficients is not required, citing literature. While this is technically correct, including them would strengthen the transparency of the analysis.

Reply: Dear reviewer, while we agree with your comments, we should highlight that Figure 2 already includes all path coefficients in the CFA. Thank you very much for allowing us to clarify this issue!

Item Generation: The authors added a detailed explanation of how items were generated, which adequately addresses the concern.

Test-Retest Reliability: The authors initially used Spearman’s correlation but revised their approach to intra-class correlation (ICC), which is more appropriate. This response is satisfactory.

Reviewer 2

Sample Size Calculation: The authors clarified that the same calculation applied to both objectives, which is a reasonable response.

Statistical Test Naming: The authors corrected the terminology to "Pearson Chi-Square," addressing the concern.

Test-Retest Analysis: The authors revised their approach to ICC, which aligns with best practices.

Table 4 Footnote: The authors added a footnote specifying the statistical test used, which improves clarity.

Reviewer 3

Prevalence Data: The authors updated the global prevalence reference but retained the 2012 Portuguese prevalence data due to the lack of newer studies. This is a reasonable decision.

Additional Questionnaires: The authors clarified the rationale for including MMSE, GDS, CES-D, and other questionnaires, which strengthens the justification.

Questionnaire Development: The authors expanded their explanation of how the questionnaire was developed based on international recommendations, which improves transparency.

Translation and Validation References: The authors added more details and references, addressing the concern.

Self-Knowledge Score Categorization: The authors provided a clear explanation of how scores were categorized, which enhances understanding.

Bias Control in Self-Reported Data: The authors acknowledged potential biases and described measures taken to mitigate them, which is a reasonable response.

Final Recommendation

The authors have adequately addressed most concerns raised by the reviewers. However, I suggest the following minor clarifications before accepting the manuscript for publication in PLOS ONE:

Additional comment 1: CFA Justification: Consider adding a brief explanation of any preliminary analysis conducted to ensure CFA was appropriate.

Reply: We have now replied to these comments (please see our reply to Reviewer 1, on this document).

Additional comment 2: Path Coefficients in CFA: While not mandatory, including them would enhance transparency.

Reply: We have now replied to these comments (please see our reply to Reviewer 1, on this document).

Additional comment 3: Bias Control: The authors could elaborate on how they ensured participants remained unbiased during questionnaire administration.

Reply: We really thank you for this suggestion. In order to better clarify this issue, we have modified and expanded the associated text (Pg 24; lines 489-497). We hope that we have adeqauetly addressed this issue.

---

## [Decision Letter · Decision Letter 3]

27 Aug 2025

PONE-D-23-34967R3Development and Validation of an Asthma Self-knowledge QuestionnairePLOS ONE

Dear Dr. Taborda-Barata,

Thank you for submitting your manuscript to PLOS ONE. After careful consideration, we feel that it has merit but does not fully meet PLOS ONE’s publication criteria as it currently stands. Therefore, we invite you to submit a revised version of the manuscript that addresses the points raised during the review process.

We look forward to receiving your revised manuscript.

Kind regards,

Luca Novelli

Academic Editor

PLOS ONE

Journal Requirements:

**Additional Editor Comments:**

Dear Authors,

Thank you for submitting the final version of your manuscript. The revisions have been carefully considered, and I am pleased to inform you that the work has been positively evaluated by the Reviewers. Your efforts to address the previous comments are much appreciated.

As a final point for improvement, as suggested by a Reviewer, I would like to draw your attention to the following:

“I recommend a final check on the statistical methodology, particularly regarding the use of Confirmatory Factor Analysis (CFA). While the authors have now included a justification and preliminary tests (KMO, Bartlett’s), it may be helpful to ensure that the rationale for bypassing a full Exploratory Factor Analysis (EFA) is sufficiently supported, given the novelty of the questionnaire. Additionally, although path coefficients are now reported in Figure 2, a brief commentary on their interpretation and relevance to the model’s robustness could further strengthen the statistical transparency. These are minor points, but addressing them would reinforce the methodological rigor of the study.”

This represents the only remaining suggestion and is considered a minor aspect".

Addressing it would further strengthen the methodological rigor of your manuscript. Thank you once again for your careful work and valuable contribution.

Reviewers' comments:

Reviewer's Responses to Questions

**Comments to the Author**

1. If the authors have adequately addressed your comments raised in a previous round of review and you feel that this manuscript is now acceptable for publication, you may indicate that here to bypass the “Comments to the Author” section, enter your conflict of interest statement in the “Confidential to Editor” section, and submit your "Accept" recommendation.

Reviewer #2: All comments have been addressed

Reviewer #4: All comments have been addressed

2. Is the manuscript technically sound, and do the data support the conclusions?

Reviewer #2: Yes

Reviewer #4: Yes

3. Has the statistical analysis been performed appropriately and rigorously? 

Reviewer #2: Yes

Reviewer #4: N/A, suggested to Editor a final check for authors about statistical methodology, particularly regarding the use of Confirmatory Factor Analysis (CFA) (see above).

4. Have the authors made all data underlying the findings in their manuscript fully available?

Reviewer #2: Yes

Reviewer #4: Yes

5. Is the manuscript presented in an intelligible fashion and written in standard English?

Reviewer #2: Yes

Reviewer #4: Yes

6. Review Comments to the Author

Reviewer #2: (No Response)

Reviewer #4: I have carefully reviewed the revised version of the manuscript submitted by the authors. I would like to commend them for the thoughtful and diligent manner in which they have addressed the comments previously raised by the reviewers, including my own.

The responses provided are clear, well-structured, and demonstrate a genuine effort to improve the quality and clarity of the work. The authors have successfully incorporated the suggested revisions, and the manuscript now presents a more robust and coherent scientific contribution.

I believe the revised version meets the standards of the journal and is suitable for publication, pending your final evaluation.

7. PLOS authors have the option to publish the peer review history of their article (what does this mean? ). If published, this will include your full peer review and any attached files.

**Do you want your identity to be public for this peer review?** For information about this choice, including consent withdrawal, please see our Privacy Policy .

Reviewer #2: No

Reviewer #4: No

---

## [Author Response · Author response to Decision Letter 4]

9 Sep 2025

Replies to Editor’s and Reviewers’ Comments to the Author

Additional Editor Comments:

Dear Authors,

Thank you for submitting the final version of your manuscript. The revisions have been carefully considered, and I am pleased to inform you that the work has been positively evaluated by the Reviewers. Your efforts to address the previous comments are much appreciated.

As a final point for improvement, as suggested by a Reviewer, I would like to draw your attention to the following:

“I recommend a final check on the statistical methodology, particularly regarding the use of Confirmatory Factor Analysis (CFA). While the authors have now included a justification and preliminary tests (KMO, Bartlett’s), it may be helpful to ensure that the rationale for bypassing a full Exploratory Factor Analysis (EFA) is sufficiently supported, given the novelty of the questionnaire.

Additionally, although path coefficients are now reported in Figure 2, a brief commentary on their interpretation and relevance to the model’s robustness could further strengthen the statistical transparency. These are minor points, but addressing them would reinforce the methodological rigor of the study.”

This represents the only remaining suggestion and is considered a minor aspect".

Addressing it would further strengthen the methodological rigor of your manuscript. Thank you once again for your careful work and valuable contribution..

Reply to the Editor:

Dear Editor,

Many thanks for your comments, which are highly constructive. We have now addressed both issues that you have raised, as follows:

a) Regarding CFA, as also suggested by Reviewer #4, we have carried a final check on the statistical methodology, and believe that the rationale for bypassing a full Exploratory Factor Analysis (EFA) is sufficiently supported, given the novelty of the questionnaire.

b) Regarding path coeficientes, we have now added a bried commnetary on their interpretation and relevance to the model’s robustness

We have, thus, added the following text sections (which include two new References):

(Pgs 7-8; lines 199-213; Materials and Methods): “Confirmatory Factor Analysis (CFA) evaluated the questionnaire's factor structure. This choice was guided by the instrument's theoretical underpinnings: each item was developed from established constructs and supported by prior literature, providing a clear a priori model to test. In situations where a predefined structure is well supported, CFA is generally preferred over Exploratory Factor Analysis (EFA), as it allows for direct testing of hypothesized relationships between observed variables and latent constructs [20].

Before carrying out the CFA, we examined the adequacy of the data for factor analysis. The Kaiser–Meyer–Olkin (KMO) statistic exceeded the recommended threshold, and Bartlett’s test of sphericity was significant, confirming that the correlation matrix was suitable for further analysis. Although the questionnaire is being applied in a new context, its strong theoretical foundation and supportive diagnostic results justified proceeding directly with CFA rather than conducting a full EFA. This approach is consistent with prior recommendations that instruments grounded in robust theory and empirical evidence may bypass EFA and move directly to CFA [21]. “

(Pag. 20; lines 403-416; Results): “Regarding observed variables → latent factors (factor loadings): The path coefficients represent standardized factor loadings, indicating how strongly each item reflects its underlying factor. High loadings (≥0.7): Items such as Q4 (0.98 on Factor 1), Q15 (0.84 on Factor 3), and Q20 (0.88 on Factor 4) are strong indicators of their factors. Moderate loadings (0.4–0.7): Items like Q1 (0.41), Q3 (0.59), and Q10 (0.63) contribute meaningfully but less strongly. Lower loadings (<0.4): Items including Q2 (0.37), Q7 (0.31), Q9 (0.32), and Q16 (0.30) are modest indicators.

Regarding latent factors ↔ latent factors (inter-factor correlations supporting construct validity): Factor 2 ↔ Factor 3 = 0.75, showing a strong relationship; Factor 2 ↔ Factor 4 = 0.73, indicating related constructs; Factor 3 ↔ Factor 4 = 0.46, a moderate relationship; and Factor 1 ↔ Factor 2 = 0.31, a modest relationship.

Regarding model implications: Some factors have items with excellent loadings (e.g., Factor 1 with Q4; Factor 4 with Q20 and Q21), making them strong measurement indicators. Overall, the model is reliable, with most items showing robust or acceptable coefficients, which boosts confidence in the validity of the factors.”

The two new references are:

20. Asparouhov T, Muthén B. Exploratory Structural Equation Modeling. Structural Equation Modeling. 2009;16(3):397–438. Doi: 10.1080/10705510903008204.

21. Marsh HW, Morin AJ, Parker PD, Kaur G. Exploratory structural equation modeling: an integration of the best features of exploratory and confirmatory factor analysis. Annu Rev Clin Psychol. 2014;10:85-110. Doi:10.1146/annurev-clinpsy-032813-153700.

---

## [Decision Letter · Decision Letter 4]

19 Sep 2025

Development and Validation of an Asthma Self-knowledge Questionnaire

PONE-D-23-34967R4

Dear Dr. Taborda-Barata,

We’re pleased to inform you that your manuscript has been judged scientifically suitable for publication and will be formally accepted for publication once it meets all outstanding technical requirements.

Kind regards,

Luca Novelli

Academic Editor

PLOS ONE

**Comments to the Author**

1. If the authors have adequately addressed your comments raised in a previous round of review and you feel that this manuscript is now acceptable for publication, you may indicate that here to bypass the “Comments to the Author” section, enter your conflict of interest statement in the “Confidential to Editor” section, and submit your "Accept" recommendation.

Reviewer #4: All comments have been addressed

2. Is the manuscript technically sound, and do the data support the conclusions?

Reviewer #4: Yes

3. Has the statistical analysis been performed appropriately and rigorously? 

Reviewer #4: Yes

4. Have the authors made all data underlying the findings in their manuscript fully available?

Reviewer #4: Yes

5. Is the manuscript presented in an intelligible fashion and written in standard English?

Reviewer #4: Yes

6. Review Comments to the Author

Reviewer #4: Dear Editor and Authors,

Thank you for your thoughtful revisions and responses to the reviewers' comments. I find that your replies adequately address the concerns raised, and the manuscript has significantly improved in clarity and quality as a result.

I appreciate the effort invested in refining the work, and I believe it is now much stronger and better positioned for publication.

7. PLOS authors have the option to publish the peer review history of their article (what does this mean? ). If published, this will include your full peer review and any attached files.

**Do you want your identity to be public for this peer review?** For information about this choice, including consent withdrawal, please see our Privacy Policy .

Reviewer #4: No

---

## [Editor Report · Acceptance letter]

PONE-D-23-34967R4

PLOS ONE

Dear Dr. Taborda-Barata,

I'm pleased to inform you that your manuscript has been deemed suitable for publication in PLOS ONE. Congratulations! Your manuscript is now being handed over to our production team.

Kind regards,

on behalf of

Dr. Luca Novelli

Academic Editor

PLOS ONE